# Dealumination and Characterization of Natural Mordenite-Rich Tuffs

**DOI:** 10.3390/ma15134654

**Published:** 2022-07-01

**Authors:** Armando Adriano, Mauricio H. Cornejo, Haci Baykara, Eduardo V. Ludeña, Joaquín L. Brito

**Affiliations:** 1Center of Nanotechnology Research and Development (CIDNA), Escuela Superior Politécnica del Litoral (ESPOL), Km 30.5 Vía Perimetral, Campus Gustavo Galindo, Guayaquil P.O. Box 09-01-5863, Ecuador; aadriano@espol.edu.ec (A.A.); hbaykara@espol.edu.ec (H.B.); popluabe@yahoo.es (E.V.L.); 2Escuela Superior Politécnica del Litoral (ESPOL), Facultad de Ingeniería Mecánica y Ciencias de la Producción (FIMCP), Km 30.5 Vía Perimetral, Campus Gustavo Galindo, Guayaquil P.O. Box 09-01-5863, Ecuador; 3Centro de Química, Instituto Venezolano de Investigaciones Científicas (IVIC), Apartado 20632, Caracas 1020-A, Venezuela; joabrito_ivic@hotmail.com; 4Biomass to Resources Group, Universidad Regional Amazónica IKIAM, Km 7, Vía Muyuna, Tena 150150, Ecuador

**Keywords:** mordenite, dealumination, catalyst support

## Abstract

The present study evaluates the feasibility of partially dealuminated natural mordenite as a catalyst support by studying improvement in its textural properties. This is the first study that reports the dealumination of natural zeolite-based tuffs from Ecuador. For this purpose, mordenite-rich tuffs were obtained from deposits close to Guayaquil, Ecuador. The raw material was micronized in order to increase its surface, and treated with NH_4_Cl. NH_4_^+^ cation-exchanged samples were finally reacted with HCl_(aq)_ to complete the dealumination process. The partially dealuminated samples were characterized using techniques such as XRD, FT-IR, SEM-EDS, and identification of their textural properties. Dealumination with HCl_(aq)_ increased the Si/Al ratio up to 9 and kept the crystallographic structure of natural mordenite, as XRD results showed that the structure of mordenite was not altered after the dealumination process. On the other hand, textural properties such as surface area and microporosity were improved as compared to natural mordenite. In view of these results, the feasibility of using natural mordenite as a catalyst support is discussed in this study.

## 1. Introduction

Zeolites are essential minerals in the chemical industry because they exist in different structures and have potential applications as adsorbents and in catalytic processes [1]. Structurally speaking, zeolites are a group of crystalline aluminosilicates whose structure contains cavities and channels; these mainly consist of tetrahedral ${\mathrm{SiO}}_{4}$ and ${\mathrm{AlO}}_{4}^{-}$ moieties, connected by oxygen bridges and charge-compensated by extra structural cations [2]. In fact, zeolites have various chemical compositions and diverse crystalline structures, which give rise to many important properties that affect adsorption and catalytic processes. Most of the studies regarding zeolites as catalytic supports deal with synthetic zeolites, due to the lack of homogeneity in their naturally occurring counterparts and the presence of various accompanying minerals [3]. However, hydrothermal treatments could tackle these issues.

Nowadays, it has been demonstrated that various hydrothermal treatments can improve the catalytic properties of zeolites [4]. Like other hydrothermal treatments, the dealumination process can also enhance the catalytic properties of zeolites by lowering the aluminum content in their structures, thus modifying their textural and acidic properties. In this sense, it is possible to increase the zeolite Si/Al ratio [5] through a hydrothermal treatment with hydrochloric or nitric acid [6,7,8]. Dealuminated mordenite, in most cases, presents changes in the porosity and acidity of its structure. Previously, some researchers have reported studies of dealuminated mordenite; they show evidence of aluminum elimination without structural change, but sometimes accompanied by a loss of crystallinity [4,9,10]. In fact, the main benefit of dealumination on mordenite is the change in porosity, which depends mainly on the kind of dealuminating agent employed and the chosen experimental conditions [4,9].

Among the zeolitic minerals, mordenite is in an important class because of its large pores, chemical resistance and thermal stability [11]. Several groups have reported on the use of mordenite as a catalyst support, aiming to produce effective catalysts. Cao et al. [12] employed alkali–acid treatment to commercial mordenites in order to improve their catalytic properties for the isomerization of ethyltoluene using Ni-Ce as the active species. No structural damage to the zeolite was found after such relatively strong chemical treatment. During their study of the transformation of soybean into biodiesel, Supamathanon and Khabuanchalad [13] prepared various potassium-loaded mordenites and optimized the catalyst properties for basic catalysis. Hu et al. [14] also used mordenite as a support for a Ru-based catalyst employed for the hydrodeoxygenation reaction of lignin and lignin-based bio-oil, aiming to improve the yield of hydrocarbons and the resistance to water of the catalyst. Fernandes et al. [15] produced a mesoporous material without structural change (pore size ranged from 30 to 50 A) via the dealumination of mordenite with hydrochloric acid. Dealumination rendered positive effects on catalyst activity and stability during the transformation of hydrocarbons. Xu et al. [16] reported the preparation of dealuminated mordenites via hydrothermal treatment, and evaluated the catalytic properties of the samples for the carbonylation of dimethyl ether. Viswanadham et al. [17] studied the dealumination of mordenite with nitric acid, and observed the increase in the surface area and volume of micropores when the concentration of acid was increased. On the other hand, Giudici et al. [18] studied the effect of nitric and oxalic acids on the dealumination of mordenite, providing evidence that oxalic acid was more effective than nitric acid.

Mordenites have also been used as adsorbents. Sawa et al. [19] investigated the effects of the dealumination of mordenite for use as an adsorbent of large-sized molecules. The authors reported that dealumination changed the porosity of mordenite when using hydrochloric acid as a dealuminating agent. Additionally, they observed that the entrances of the pores became enlarged at the optimum temperature of 358 K; thus, the samples dealuminated at 358 K were adequate for trimethylpentane adsorption, while the results for the samples dealuminated above that temperature were adverse for this application.

All these reports illustrate the ample range of possible uses for this zeolite, and demonstrate significant changes in the microporosity of mordenite without modification of the crystal structure upon application of diverse dealumination protocols. It is estimated that the cavities left by the extracted aluminum atoms are filled in by hydroxyl groups bonded to silicon atoms [20]. Therefore, the dealumination process can be used to control the composition of Al and Si atoms in the framework of mordenite, increase surface area, and decrease the number of acid sites. In summary, the structural and compositional changes observed in mordenite after dealumination treatments can transform the dealuminated mordenite into a catalytic support or adsorbent for various industrial processes [21].

Despite the importance of the dealumination of zeolites for improving the properties of catalytic materials, the chemical modification of natural mordenite, one of the most abundant zeolites worldwide, has been comparatively little studied. Kurniawan et al. [22] compared a natural Indonesian mordenite with a synthetic faujasite for the production of bio-hydrocarbon oils from coconut soap. Meanwhile Fani and co-workers investigated the valorization of a natural Greek mordenite for use as support in a Ni catalyst to upgrade biodiesel into renewable diesel [23]. These very recent reports demonstrate the interest in developing high-technology materials from natural sources.

For the economy of a developing country such as Ecuador, exploitation of its natural resources is a necessity. Thus, the present work’s main goal is to obtain and analyze the textural and structural properties of dealuminated, natural Ecuadorian mordenite, aiming to produce useful materials for catalytic and adsorption processes. After being dealuminated by using hydrochloric acid and ammonium chloride, the resulting materials were characterized via X-ray diffractometry (XRD), Fourier transform infrared spectroscopy (FTIR), and scanning electron microscopy–energy-dispersive X-ray spectroscopy (SEM-EDS). Additionally, the textural properties of the samples were determined via nitrogen physisorption. The results and implications of the dealuminated mordenite properties are also analyzed.

## 2. Experimental Procedure

### 2.1. Raw Material

The raw material, mordenite-rich tuff, was provided by ZEOLITA S.A., Guayaquil (Ecuador). Mordenite is present in the Cayo formation, west of Guayaquil [24]. Ammonium chloride (NH_4_Cl, Fisher Chemical, Hampton, NH, USA) and hydrochloric acid (HCl_(aq)_, 37%) were used as received. The raw material was micronized to a grain size lower than 15 µm and then washed with distilled water and hydrogen peroxide. The washed, and later, dried raw material was identified as the ZL sample.

### 2.2. Dealumination

The dealumination of mordenite was carried out according to the procedure reported by Garcia-Basabe et al. [5]. The mordenite NH_4_^+^ cation-exchanged form (ZAM) was obtained by refluxing a portion of ZL sample with 1 mol/L of ammonium chloride solution. Afterwards, 0.6 M hydrochloric acid was used for dealumination for 1, 3, and 5 consecutive cycles [5]. Hence, those samples were labeled as ZDES1, ZDES3, and ZDES5 according to the number of treatment cycles.

### 2.3. Characterization

All the samples were dried in an oven at 60 °C for 48 h before applying the characterization techniques. Particle size distribution of the ZL sample was obtained using a particle size analyzer, the Masterizer 2000 Scirocco (Malvern Panalytical Ltd., Eindhoven, The Netherlands), using the dry dispersion method.

The diffraction patterns were collected on a PANalytical X’Pert PRO X-ray diffractometer (Panalytical Ltd., Eindhoven, The Netherlands) employing Cu K ∝ radiation. The samples were analyzed in a 2θ range of 5–70°, with a step size of 0.017° and a scanning time of 7 s per step. High Score Plus^®^ software (Malvern Panalytical Ltd., Eindhoven, The Netherlands) was used to calculate crystallographic parameters using the Rietveld Refinement method in order to analyze the diffraction patterns [25]. ZnO was used as an internal standard. The procedure of the Rietveld method reported by Snellings et al. was used in order to quantify the mineral composition of raw material and of the corresponding dealuminated samples [26].

The morphologies of the samples were studied using the scanning electron microscopy technique by means of the FEI Inspect S50 instrument (Thermo Fisher, Waltham, MA, USA), with a voltage of 7.5 kV in high vacuum mode, under a pressure of 35–40 Pa. Elemental analysis was achieved using the energy dispersive X-ray spectroscopy method, with a voltage of 20 kV in high vacuum mode. For this purpose, the standard error was estimated using the average of several repeated measures under the same conditions [27]. Standard deviations were calculated according to the study reported by Streiner [28].

FTIR spectra were collected using an FTIR Spectrum 1000 spectrophotometer (PelkinElmer, Waltham, MA, USA). Each run was measured as the average of 12 scans with a resolution of 32 cm^−1^, in the range of 4000–400 cm^−1^, with a purge of dry nitrogen (1 ft^3^/min). All samples were prepared using the KBr pellet method.

The textural properties were studied using a Micromeritics ASAP 2010 instrument via nitrogen physisorption at −196 °C. Before measurements were taken, samples were outgassed under vacuum at a temperature slightly above 100 °C.

## 3. Results and Discussion

Figure 1 shows the particle size distribution of the as-received raw material, mordenite-rich tuff. As seen in the figure, approximately 90% of the volume of the particles has a size lower than 13.6 mm.

The results obtained using the X-ray diffraction method indicate that the zeolite-rich tuff used in the present study mainly contains mordenite (32%; ISCD 029-1257), quartz (23%; ICSD 033-1162), and amorphous phases (45%). Figure 2 shows the X-ray diffraction patterns of the samples. Generally, the intensity of mordenite peaks corresponding to the Miller indices (110), (200), (111), (400), (241), (202), (511), and (620) decrease with increasing intensity of the hydrothermal treatment. This is obvious evidence of the impact of hydrochloric acid treatment on the raw material in the dealumination process. The main effect of the dealumination treatment is the decrease in the crystallinity of mordenite (see Figure 2). Similar results were obtained by other authors who carried out the dealumination process of mordenite using HCl_(aq)_ [7,29,30].

For crystallinity analysis, peak profile fitting was carried out using High Score Plus^®^ software, thus obtaining peak widths at full-width at half-maximum (FWHM). The decreases in crystallinity of all the samples were analyzed via differences between the peak width $H_{w}$ (FWMH) values. Figure 3 reveals no difference between the ZAM and ZDES1 samples, which means that there is no change regarding the crystallinity of mordenite after a single cycle of dealumination. The scenario is somewhat different when analyzing the changes in H_w_ between ZDES1-ZDES3 and ZDES3-ZDES5 samples, as the values obtained reveal slight changes in the crystallinity of mordenite samples. Nevertheless, the results shown in Table 1 provide evidence that the structure of mordenite is preserved after the cyclic dealumination process. Narayanan et al. [30], González et al. [21], and Ha et al. [20] reported similar results after the dealumination of mordenite by hydrochloric acid, which reveals the resilience of the MOR structure to relatively strong hydrotreatment processes.

Figure 4 shows the microstructures of the ZL, ZAM, and ZDES1 samples. As seen in the figure, there are some crystalline structures that belong to the mordenite and quartz phases, as identified by XRD. It is evident that there are not any significant changes in the crystalline structures of the minerals in the treated and untreated samples.

The elemental analysis of dealuminated mordenite is presented in Table 2. The obtained results show the variation in the chemical composition of mordenite after the dealumination process. After the treatment with NH_4_Cl solution and following various dealumination cycles, the amounts (wt.%) of the cations, i.e., ${\mathrm{Na}}^{+}$, $K^{+}$, ${\mathrm{Ca}}^{2+}$ and ${\mathrm{Mg}}^{2+}$ were significantly decreased (see Table 2). This is probably due to the exchange of such cations by $H^{+}$ after NH_4_^+^ removal and the acid treatment [20]. In addition, there is clear evidence that the Si/Al ratio increases upon application of the dealumination cycles (see Table 2 and Figure 5). Detailed SEM-EDS images of all samples can be seen in Appendix A. While the low magnification of these micrographs does not allow us to find significant differences between the morphologies of the dealuminated samples, the presence of non-zeolitic phases can be appreciated for the raw material, especially the ~10 wt.% content of metallic elements that are distinct from Si and Al. 

These results indicate that the structure of mordenite loses aluminum atoms, leaving vacant sites (structure defects) that probably become stabilized by the formation of hydroxyl nests [31]. As seen in Figure 5, ZL and ZAM samples show similar Si/Al values, which means that the ZAM structure was not affected by the cation exchange process with NH_4_Cl_(aq)_. On the other hand, in the first dealumination cycle, ZDES1, the Si/Al ratio increased to approximately 5.83, suggesting dealumination of the zeolite structure (however, see below). Samples ZDES3 and ZDES5 show 6.47 and 9.38 Si/Al ratios, respectively. The increase in the Si/Al ratio becomes steeper for ZDES5.

This is probably due to aluminum remnants being obstructed or blocked in the porous system [32,33]. González et al. [21] and Sawa et al. [19] dealuminated mordenite using a hydrochloric acid treatment method and obtained high Si/Al values.

On the other hand, while most metallic impurities are essentially diminished to similar low contents for all samples after the first dealumination cycle, the amount of Fe, instead, decreases gradually after each cycle of dealumination. The reason for this behavior of Fe is probably related to this element partially occupying three different types of sites in natural zeolites: [34] as an Al substitute in the structure; as a cation; or as an impurity.

Figure 6 illustrates the FTIR spectra of the samples in the range between 1750 and 500 cm^−1^. The most notable change for dealumination is evidenced by the increase in the frequencies of asymmetric T–O (T = Al or Si) stretching peaks around 1041.8 cm^−1^. Those peaks shifted to higher frequencies for each treatment cycle compared to the raw material, ZL. As seen in Figure 6, the ZDES5 spectrum shows a peak at 1062.3 cm^−1^, which is the highest shift as compared to ZL among all the samples. Hence, the treatment cycles and shift in the peak indicate that the substitution of Al atoms by Si atoms in the structure of mordenite takes place [3,35]. Therefore, the increase in the frequency is related to the increase in the Si/Al ratio [10]. Furthermore, the symmetrical stretching peak observed at 785.8 cm^−1^ also slightly increases with dealumination.

González et al. [21] reported high values of frequencies for symmetrical and asymmetric T–O stretching, which they used to show evidence of the removal of aluminum from the mordenite. The peak around 1400.9 cm^−1^ is assigned to the vibrational mode of ${\mathrm{NH}}_{4}^{+}$. Since Si atoms are more electronegative Si-O bond is shorter than Al-O bond. Due to this, the frequency of Si-O vibrations shows at a higher frequency. This result is consistent with the literature [21,36], which appears in the ZAM and ZDES1 samples.

On the other hand, the vibrations in the region of 1600–4000 cm^−1^ correspond to the presence of zeolitic water [37].

Figure 7 shows peak intensities around 3437 cm^−1^, which reveals the formation of silanol nest defects in vacant Al sites left by dealumination [5,38]. That band belongs to the Si-OH group bound to hydrogen in nest defects, and contributes to hydrogen bonds supported by free and crystalline water molecules.

Nitrogen physisorption isotherms obtained for all samples (not shown) are of type IV in the BDDT classification adopted by IUPAC. They present hysteresis loops of type H3 in the IUPAC classification, which approximately reach relative pressures between 0.4 and 1. Although hysteresis loops in this region generally reflect mesopores of large size, this specific type of loop, rather, corresponds to pores of slit type due to the kind of mineral crystalline structure.

Textural data derived from the N_2_ physisorption measurements are shown in Table 3. According to these results, the ZL sample shows a low surface area, less than 10 m^2^/g, while the pore volume is close to 0.05 cm^3^/g. The micropore surface area and volume are also quite low. García Basabe et al. [5] obtained similar low values of textural properties with their samples of natural zeolite.

In the ZAM sample, cationic exchange with ammonia results a decrease in area and pore volume. This occurred due to the blockage of the porous system by ammonium cations [15], without contribution from micro-pores. The ZDES1 sample, in turn, increases its pore volume, resulting in values of the average pore diameter like those of the ZL sample. This demonstrates the unblocking of the porous system as a result of the removal of ammonium cations. However, the BET surface area remains at a low value, similar to that of ZAM, after just a single dealumination cycle. This means that treatment of the ammonia-exchanged sample using aqueous HCl (first cycle) is unable to remove any Al from the zeolite structure, likely due to some protective effect of the corresponding surface species by the ammonia cations. The decreased Al content after the first cycle is probably due to the removal of only extra-structural aluminum. In further cycles, aluminum from the zeolitic structure would be removed, which explains the increased textural properties. As in the previous scenario, for the ZDES1 sample, there is no contribution from micropores to the textural properties.

The results for the ZDES3 sample provide evidence for a significant increase in the BET area, although the porous volume decreases, as do the average pore diameter values. On the other hand, the BET area of ZDES5 strongly increases, with a significant contribution (>50%) from the micropores present in the material; the pore volume is lower than that of the ZL sample, but it is the highest value among the chemically modified samples, receiving a contribution of microporous volume close to 25% of the total. The average pore diameters decrease, evidently due to the contribution of micropores to the textural properties of the material.

In Figure 8 the distributions of the pore diameters of all samples can be seen, essentially showing that most pores measure around 40 nm (the maximum in all the distribution curves). The average diameter is larger due to the presence of macropores and intra-particle volume. Sawa et al. [19] and Fernandes et al. [15] obtained pores of diameter between 5–50 nm and 3–3 nm after the dealumination of mordenite.

The results provide evidence that the dealumination cycles contributed to the improvement of the textural properties of natural mordenite. The chemical treatment resulted in the development of an important surface area, as well as micro-porosity. The latter is not particularly beneficial in the context of the hydrotreatment of oil and its fractions, one of the intended applications of this natural material in Ecuador; this is because the molecules most refractory to this process tend to be relatively large and require a considerable mesoporous fraction [39,40,41]. However, other possible applications in catalysis can be explored, and these materials could be useful for the adsorption of ethylene, in the context of fruit conservation for exports [42]. Such applications are currently under evaluation.

## 4. Conclusions

The dealumination of mordenite using a procedure involving cycles of acid hydrothermal treatment gave rise to the improvement of textural properties, including an increase in the surface area and micro-porosity of the zeolite. The ZDES5 sample, obtained after five cycles of acid treatment, showed an important increase in the BET area, with a significant contribution (>50%) from the micropores present in the material. In addition, the greatest Si/Al ratio of 9.38 was obtained in the highest treated sample, ZDES5. The removal of aluminum atoms from mordenite did not cause a significant structural change, as retention of the crystalline structure of mordenite was verified by XRD. Still, there was a slight change in crystallinity that could be evidenced quantitatively by the differences between the values of the peak widths (FWHM) of the samples. Interestingly, the first dealumination cycle removed only extra-structural Al, and a certain amount of Fe seems to be associated with the zeolitic phase. Further studies about the feasibility of dealuminated mordenite samples as adsorbent or catalyst supports are under consideration.

## Figures and Tables

**Figure 1 materials-15-04654-f001:**
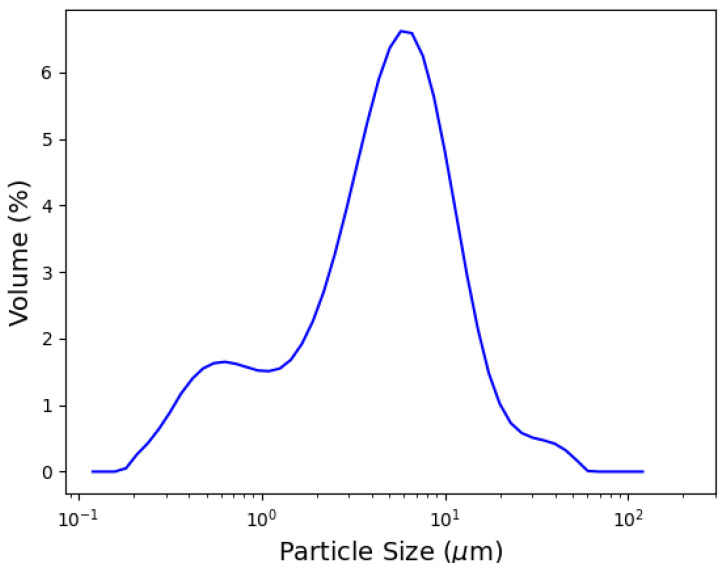
Particle size distribution of the as-received raw material. d(0.1) = 0.7 µm, d(0.5) = 5.1 µm, and d(0.9) = 13.6 µm.

**Figure 2 materials-15-04654-f002:**
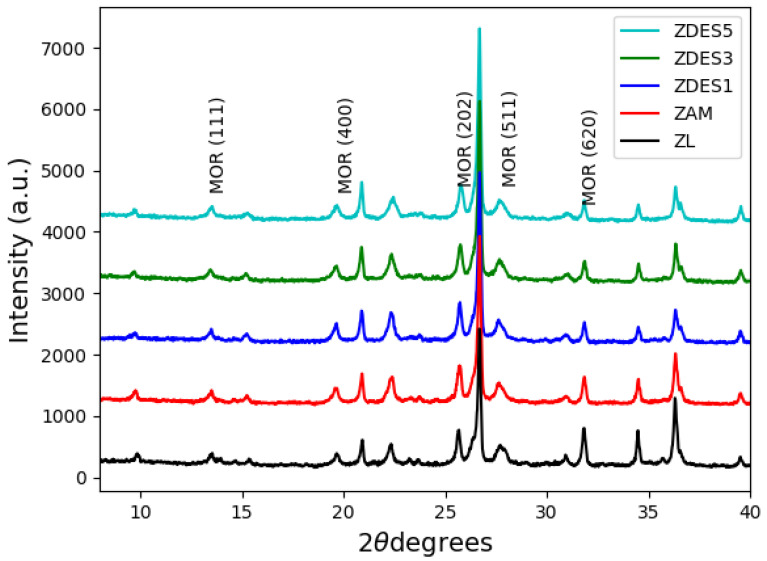
XRD patterns of the raw material (ZL); ammonia-exchanged sample (ZAM); and dealuminated samples (ZDES1, ZDES3, and ZDES5).

**Figure 3 materials-15-04654-f003:**
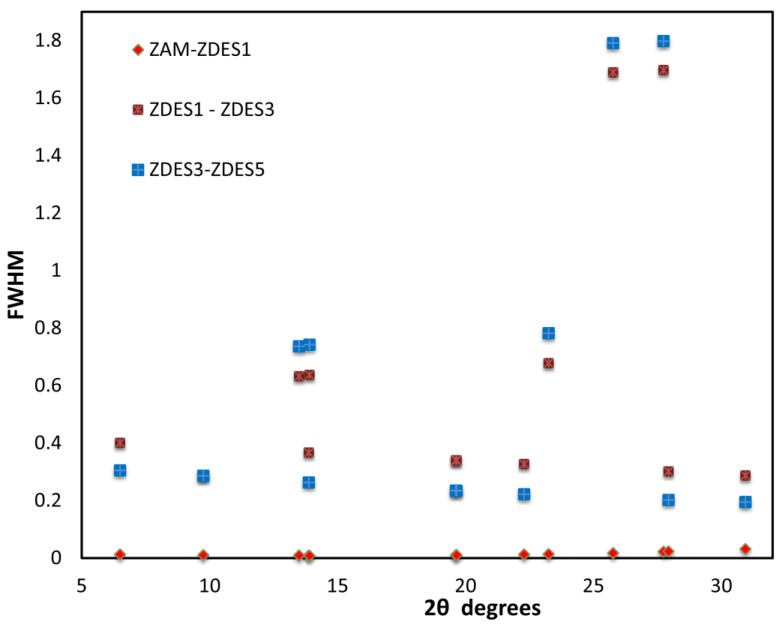
Differences in peak width $H_{w}$ versus 2θ of the raw material (ZL); ammonia-exchanged sample (ZAM); and dealuminated samples (ZDES1, ZDES3, and ZDES5).

**Figure 4 materials-15-04654-f004:**
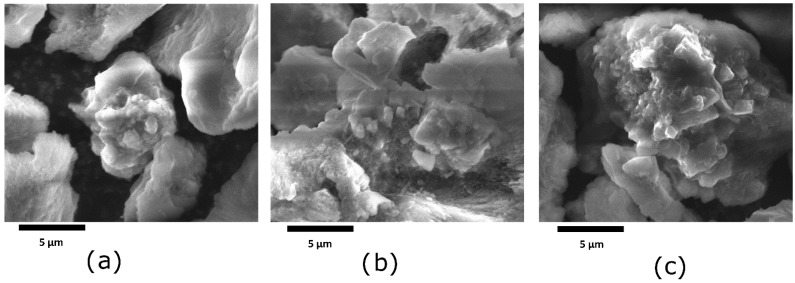
Microstructures of: (**a**) raw material (ZL); (**b**) ammonia-exchanged sample (ZAM); and (**c**) dealuminated sample for 1 cycle (ZDES1).

**Figure 5 materials-15-04654-f005:**
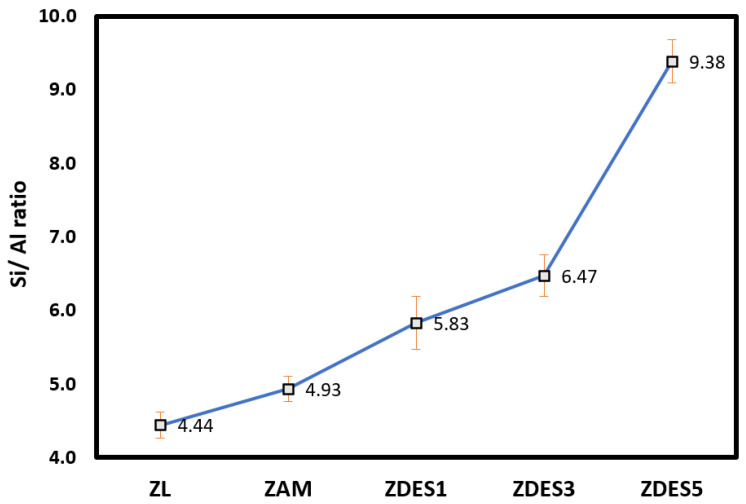
Variation in Si/Al ratio of the raw material (ZL); ammonia-exchanged sample (ZAM); and dealuminated samples (ZDES1, ZDES3, and ZDES5).

**Figure 6 materials-15-04654-f006:**
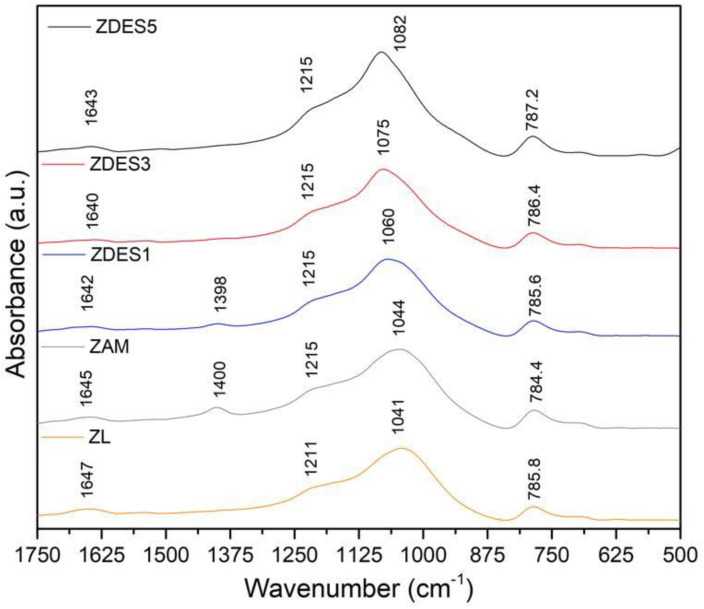
FTIR spectra in the range of 1750–500 cm^−1^ of the raw material (ZL); ammonia-exchanged sample (ZAM); and dealuminated samples (ZDES1, ZDES3, and ZDES5).

**Figure 7 materials-15-04654-f007:**
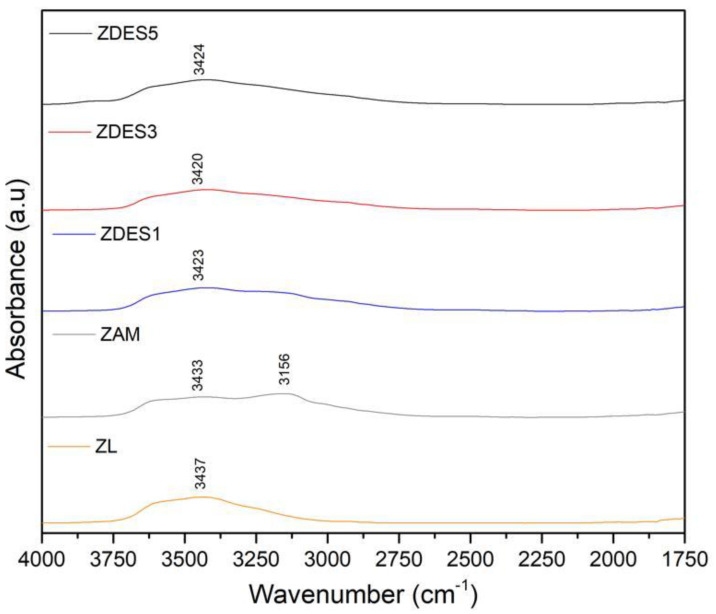
FTIR spectra in the range of 4000-1750 cm^−1^ of the raw material (ZL); ammonia-exchanged sample (ZAM); and dealuminated samples (ZDES1, ZDES3, and ZDES5).

**Figure 8 materials-15-04654-f008:**
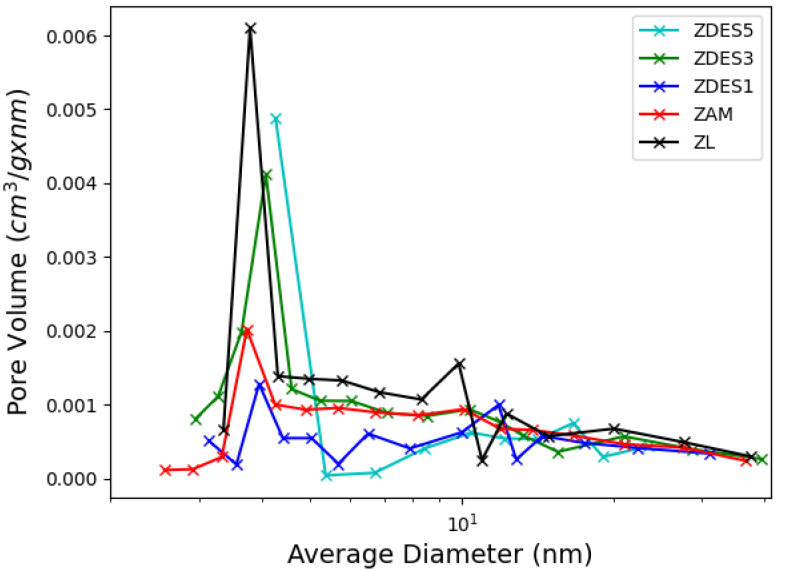
Pore diameter distribution of samples.

**Table 1 materials-15-04654-t001:** Unit-cell parameters and volume of the raw material (ZL); ammonia-exchanged sample (ZAM); and dealuminated samples (ZDES1, ZDES3, and ZDES5) *.

Samples	a (Å)	b (Å)	c (Å)	α, β, γ (^ο^)	Volume* (Å^3^)
ZL	18.042 (1)	20.397 (2)	7.481 (2)	90	2775.19
ZAM	18.143 (2)	20.384 (3)	7.500 (1)	90	2774.10
ZDES1	18.182 (2)	20.371 (1)	7.492 (3)	90	2775.12
ZDES3	18.181 (1)	20.382 (2)	7.494 (2)	90	2777.18
ZDES5	18.161 (3)	20.322 (2)	7.473 (4)	90	2758.20

* Unit-cell and volume parameter values were obtained via refinement (Rietveld method) with mordenite, ISCD code 029-1257.

**Table 2 materials-15-04654-t002:** Elemental chemical analysis of: raw material (ZL), ammonia-exchanged form (ZAM), and dealuminated samples (ZDES1, ZDES3, and ZDES5). The values in parenthesis are the standard deviation error of each element.

wt.%	ZL	ZAM	ZDES1	ZDES3	ZDES5
O	42.70 (0.61)	42.87 (0.71)	43.93 (0.61)	43.88 (1.03)	43.81 (1.37)
Al	9.15 (0.30)	9.34 (0.26)	8.19 (0.35)	7.42 (0.22)	5.38 (0.21)
Si	40.21 (0.92)	45.70 (0.69)	46.59 (0.77)	47.49 (1.08)	49.99 (1.19)
Na	1.18 (0.33)	0.07 (0.01)	0.06 (0.01)	0.05 (0.01)	0.05 (0.01)
K	1.96 (0.78)	0.42 (0.06)	0.38 (0.08)	0.39 (0.11)	0.25 (0.05)
Ca	2.88 (0.52)	0.14 (0.01)	0.12 (0.01)	0.08 (0.01)	0.11 (0.02)
Mg	0.61 (0.15)	0.25 (0.05)	0.17 (0.03)	0.18 (0.04)	0.12 (0.01)
Fe	1.32 (0.33)	1.22 (0.60)	0.56 (0.13)	0.50 (0.17)	0.30 (0.05)

**Table 3 materials-15-04654-t003:** Textural properties of the raw material (ZL); ammonia-exchanged sample (ZAM); and dealuminated samples (ZDES1, ZDES3, and ZDES5).

Sample	BET Area (m^2^/g)	Microporous Area (m^2^/g)	Porous Volume (cm^3^/g)	Microporous Volume (cm^3^/g)	Average Pore Diameter (Å)		
					4Vp/A	BJH Ads	BJH Des
					*	**	***
ZL	9.7	0.2	0.055	0.0001	226.79	303.44	157.79
ZAM	6.21	-	0.0288	-	185.63	185.98	127.89
ZDES1	6.65	0.01	0.0402	-	242.11	288.68	162.97
ZDES3	10.52	-	0.032	-	121.73	152.57	124.41
ZDES5	36.94	20.46	0.0442	0.091	47.85	169.69	128.08

* Gurvitsch’s rule; ** and *** values obtained from BJH adsorption and desorption, respectively.

## Data Availability

Not applicable.

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
