# Peer review of "Dealumination and Characterization of Natural Mordenite-Rich Tuffs"

_materials, 2022, doi:10.3390/ma15134654_

Round 1
Reviewer 1 Report
The manuscript also must be completely checked and reorganized. Example: The authors did not follow basic rules of MDPI about preparation of manuscript since your citations in the text are not given in adequate way. According to the MDPI rules references in the text should be given numerically but not by the surname of the first authors.
Abstract was relatively broad and did not point out the main findings and the novelty of the work. Abstract must be enriched via valuable results which pave the way for understanding the audiences.
Some important RECENT research results in this area should be discussed and cited in the section of introduction, so that, on the one hand we can provide a solid background and progress to the readers regarding the current state-of-knowledge on this topic, and on the other hand you also can tell readers what the highlights of your manuscript are compared with previous progresses about natural modernite and its potential applications.
Materials and methods. The methodology section is not well organized for the readers to understand the concept. Please describe briefly equipment used in the experiment – work development environment / work apparatus should be given – model of equipment (manufacturer, city, country). Please detail the determination the composition of the zeolite-rich tuff: “The results show that zeolite-rich tuff is composed of mordenite (32%), quartz (23%), and amorphous (45%) content”. Please verify again the XRD measurement conditions.
Table 1. How was determined the standard deviation error of each element?
Results and discussion section must be considerably improved/ more technically presented. All the figures should be arranged in proper sequence and figure caption should be elaborated. All the results obtained should be compared with the reports (preferably recent) in the literature. More care should be taken to present the results and facilitate understanding of the work.
“Detailed SEM-EDS images of all samples can be seen in Figures S1-S5 in Supporting information File”? The supporting Information file is missing. Please introduce in the manuscript the mentioned SEM, TGA-DSC, FT-IR and the corresponding discussions.
In the tables, for decimal point, “dots” should be used instead for “commas”.”.
Please merge Figures 7-11.
The manuscript is full of grammatical and writing (flow) mistakes. So, the language is not suitable for publication in a scientific journal in many parts. If it possible, I recommend that the authors ask for the help of a native speaker scientist/colleague, who can help with English grammar and scientific writing style. Also, please replace the “Spanish” words.
Finally, I consider that the paper is not proper for publication in the present format and requires large, large revision. I hope that in the near future all the issues will be solved. If the manuscript will not be considerable improved, I will not recommend its publication.
Author Response
Comments and Suggestions for Authors
- The manuscript also must be completely checked and reorganized. Example: The authors did not follow basic rules of MDPI about preparation of manuscript since your citations in the text are not given in adequate way. According to the MDPI rules references in the text should be given numerically but not by the surname of the first authors.
Response: Thank you for your comment. The reference style of the manuscript has been changed according to the reference style of Materials.
- Abstract was relatively broad and did not point out the main findings and the novelty of the work. Abstract must be enriched via valuable results which pave the way for understanding the audiences.
Response: Thank you for your comment. The abstract section has been improved.
- Some important RECENT research results in this area should be discussed and cited in the section of introduction, so that, on the one hand we can provide a solid background and progress to the readers regarding the current state-of-knowledge on this topic, and on the other hand you also can tell readers what the highlights of your manuscript are compared with previous progresses about natural modernite and its potential applications.
Response: The introduction and Results & Discutions have been improved according to reviewer's suggestions
- Materials and methods. The methodology section is not well organized for the readers to understand the concept. Please describe briefly equipment used in the experiment – work development environment / work apparatus should be given – model of equipment (manufacturer, city, country). Please detail the determination the composition of the zeolite-rich tuff: “The results show that zeolite-rich tuff is composed of mordenite (32%), quartz (23%), and amorphous (45%) content”. Please verify again the XRD measurement conditions.
Response: We agree with the reviewer. The following part about the details of XRD analysis has been given under 2.3. Characterization subsection: “High Score Plus® software was used to calculate crystallographic parameters by the Rietveld Refinement method to analyze the diffraction patterns [19]. In this case, ZnO was used as an internal standard. The procedure of the Rietveld method reported by Snellings et al. was used in order to quantify the mineral composition of raw material and corresponding dealuminated samples [20].”
- Table 1. How was determined the standard deviation error of each element?
Response: Thank you for your comment. The following sentence has been added to the text: “Standard deviations given in Table 1 have been calculated according to the study reported by Streiner [19].”
- Results and discussion section must be considerably improved/ more technically presented. All the figures should be arranged in proper sequence and figure caption should be elaborated. All the results obtained should be compared with the reports (preferably recent) in the literature. More care should be taken to present the results and facilitate understanding of the work.
Response: Thank you for your comment. The necessary corrections and improvements have been done in the entire the text.
- “Detailed SEM-EDS images of all samples can be seen in Figures S1-S5 in Supporting information File”? The supporting Information file is missing. Please introduce in the manuscript the mentioned SEM, TGA-DSC, FT-IR and the corresponding discussions.
Response: We are sorry for the inconvenience. We have not uploaded to upload the supporting information file mistakenly while we have uploaded all the other files. Supporting information file has been uploaded this time.
- In the tables, for decimal point, “dots” should be used instead for “commas”.”.
Response: Thank you for your comment. The necessary corrections have been done in all the tables.
- Please merge Figures 7-11.
Response: The figures have been merged.
- The manuscript is full of grammatical and writing (flow) mistakes. So, the language is not suitable for publication in a scientific journal in many parts. If it possible, I recommend that the authors ask for the help of a native speaker scientist/colleague, who can help with English grammar and scientific writing style. Also, please replace the “Spanish” words.
Response: We are sorry for the the inconvenient. The manuscript has been revised and improved. All the Spanish words have been deleted and necessary correction were done. We appreciate the reviewer for this valuable observation.
- Finally, I consider that the paper is not proper for publication in the present format and requires large, large revision. I hope that in the near future all the issues will be solved. If the manuscript will not be considerable improved, I will not recommend its publication.
Response: We appreciate the reviewer for the comments. The manuscript has been improved thanks to the comments and suggestions of all the reviewers.

Reviewer 2 Report
(a) The manuscript should be thoroughly revised in language.
(b) Redraw Fig. 1 using more sophisticated software.
(c) Redraw Fig. 3 with more distinctive labels.
(d) Remove the digital labels in Fig. 4.
(e) Redraw Figs. 7-11 by changing the x-axis to English. Reorganize the figures in a representative way, instead of simply presenting all data.
Author Response
Dear editor,
I am submitting the revised version of our manuscript titled “Dealumination of Natural Mordenite and Its Potential Use as Catalyst Support” for consideration for publication in Materials.
Comments and Suggestions for Authors
- The manuscript should be thoroughly revised in language.
Response: Thank you for your comment. The manuscript has been thoroughly revised and improved accordingly.
- Redraw Fig. 1 using more sophisticated software.
Response: Thank you so much for your comment. Figure 1 is the figure given by the software. The image will be the same even if it is drawn by any other software, therefore we prefer to keep the figure as it is.
- Redraw Fig. 3 with more distinctive labels.
Response: Thank you so much for your comment. Figures 2 and 3 have been updated.
- Remove the digital labels in Fig. 4.
Response: Thank you for your comment. Since the digital labels will not be active in pdf, we prefer to keep the figure in its current form. Thank you for your understanding.
- Redraw Figs. 7-11 by changing the x-axis to English. Reorganize the figures in a representative way, instead of simply presenting all data.
Response: The caption of Figures 7-11 has been corrected and they are presented in a unique figure (figure 7).

Reviewer 3 Report
The present manuscript needs substantial improvement. Most of the figures have poor presentation, including that they are written in Spanish. The language quality is poor and needs to be corrected. Moreover, the novelty of the work is not clear, and the literature discussed in the introduction is not enough discussed and updated. Major results e.g TG-DTA and SEM-EDS that are described in methods are not included, and the results included are not enough to support the characterization of the materials. Based on this, the recommendation of the manuscript in the present form is not to be published.
- The literature review in the introduction is not extensive and out of date, citing works only until 2017. There are more recent works on the use of mordenite as catalyst support:
https://doi.org/10.1021/acsomega.1c02809
https://doi.org/10.1016/j.matpr.2019.06.162
https://doi.org/10.1016/j.joei.2021.03.017
- There are some typos e.g. “row” (Line 74, Page 2). It is recommendable to do a language improvement in the manuscript.
- In the XRD figure, please include the peaks of the card in the figure to facilitate the identification. Also, the miller index of the mordenite are discussed, but there are several peaks after 30 degrees not identified. The x and y axis of this figure are written in Spanish.
- The Figures 7, 8, and 10 are also written in Spanish.
- TG-DTA and SEM-EDS analysis are described in experimental section but the results are not included.
Author Response
Dear editor,
I am submitting the revised version of our manuscript titled “Dealumination of Natural Mordenite and Its Potential Use as Catalyst Support” for consideration for publication in Materials.
Comments and Suggestions for Authors
The present manuscript needs substantial improvement. Most of the figures have poor presentation, including that they are written in Spanish. The language quality is poor and needs to be corrected. Moreover, the novelty of the work is not clear, and the literature discussed in the introduction is not enough discussed and updated. Major results e.g TG-DTA and SEM-EDS that are described in methods are not included, and the results included are not enough to support the characterization of the materials. Based on this, the recommendation of the manuscript in the present form is not to be published.
- The literature review in the introduction is not extensive and out of date, citing works only until 2017. There are more recent works on the use of mordenite as catalyst support:
https://doi.org/10.1021/acsomega.1c02809
https://doi.org/10.1016/j.matpr.2019.06.162
https://doi.org/10.1016/j.joei.2021.03.017
Response: Thank for your comment. The suggested references have been added to the introduction section as follows: “Cao et al. [11] improved catalytic properties of commercial mordenites by alkali-acid treatment without any stuructural damage. Supamathanon and Khabuanchalad [12] prepared various potassium loaded mordenites to be used as catalyst for the transformation of soybean oil into biodiesel. Hu et al. [13] used mordenite as a support for a Ru-based catalyst used for hydrodeoxygenation reaction of lignin and lignin-based bio-oil.”
- There are some typos e.g. “row” (Line 74, Page 2). It is recommendable to do a language improvement in the manuscript.
Response: We appreciate the reviewer for the observations. We have revised the entire document.
- In the XRD figure, please include the peaks of the card in the figure to facilitate the identification. Also, the miller index of the mordenite are discussed, but there are several peaks after 30 degrees not identified. The x and y axis of this figure are written in Spanish.
Response: Thank you so much for your comment. The XRD figure (Figure 2) has been improved and Spanish words were eliminated.
- The Figures 7, 8, and 10 are also written in Spanish.
Response: Thank you for your comment. The figures mentioned have been corrected and combined in a single figure (Figure 7).
- TG-DTA and SEM-EDS analysis are described in experimental section but the results are not included.
Response: We are so sorry for the inconvenient. We have mistakenly mentioned TGA-DTA in the manuscript. Due to that TGA-DSC words have been removed from the text.

Reviewer 4 Report
This manuscript is a detailed study which deals with the strategy to dealuminate micronized mordenite with NH4Cl and HCl. The authors investigated the potential use of partially dealuminated natural mordenite as catalyst support. Overall, the logic is unclear; some results need to be discussed in-depth. I hope the below comments will be able to help to improve this work further.
Major issue:
The main shortcoming of the manuscript is;
- The manuscript would benefit from thorough proofing for language. The overall writing is understandable, but there are significant phrasing and stylistic issues that impair the quality of the manuscript and undermine its other qualities.
Minor issue:
- Be careful about citing old references. The rule of thumb is to go back at most five to six years.
- Since the study of dealuminated natural mordenite sample as catalyst support hasn't been carried out. Please consider revising the title.
- Page 2 Line 79, Please define 'ZL'.
- Page 2 Line 85, Please define 'ZAM'.
- Figs 5 and 6, please include y-axis.
- Why not just combine Figs 5 and 6?
- Please show the method of how the author determined FWMH.
Author Response
Dear editor,
I am submitting the revised version of our manuscript titled “Dealumination of Natural Mordenite and Its Potential Use as Catalyst Support” for consideration for publication in Materials.
Comments and Suggestions for Authors
This manuscript is a detailed study which deals with the strategy to dealuminate micronized mordenite with NH4Cl and HCl. The authors investigated the potential use of partially dealuminated natural mordenite as catalyst support. Overall, the logic is unclear; some results need to be discussed in-depth. I hope the below comments will be able to help to improve this work further.
Major issue:
The main shortcoming of the manuscript is;
- The manuscript would benefit from thorough proofing for language. The overall writing is understandable, but there are significant phrasing and stylistic issues that impair the quality of the manuscript and undermine its other qualities.
Response: The entire the manuscript has been revised and improved. We thank the reviewer for very valuable comments and suggestions.
Minor issue:
- Be careful about citing old references. The rule of thumb is to go back at most five to six years.
Response: Thank you for your comment. We have cited some new references.
- Since the study of dealuminated natural mordenite sample as catalyst support hasn't been carried out. Please consider revising the title.
Response: Thank you for your comment. We agree with the reviewer. The title has been changed to “Dealumination and characterization of Natural Mordenite-Rich Tuffs.”
- Page 2 Line 79, Please define 'ZL'.
Response: Thank you for your comment. The definition of ZL has been given in the manuscript.
- Page 2 Line 85, Please define 'ZAM'.
Response: Thank you for your comment. The definition of ZAM has been given in the manuscript.
- Figs 5 and 6, please include y-axis.
Response: The “absorbance (a.u)” given in the figures is the definition of y-axis.
- Why not just combine Figs 5 and 6?
Response: We appreciate the reviewer for the comment. We have presented those figures separated to show the differences and for a better resolution. Therefore, we prefer to keep the figures as separated.
- Please show the method of how the author determined FWMH.
Response: Thank you for your comment. The following sentence has been added in the manuscript. “For crystallinity analysis, a peak profile fitting was carried out with High Score Plus software, thus obtaining peak widths at full width at half maximum (FWHM).”

Round 2
Reviewer 1 Report
This manuscript was somewhat improved after the first submission, but critical mistakes remain present.
Results and discussion section must be considerably improved/ more technically presented. All the figures should be arranged in proper sequence and figure caption should be elaborated. All the results obtained should be compared with the reports (preferably recent) in the literature. More care should be taken to present the results and facilitate understanding of the work.
“Detailed SEM-EDS images of all samples can be seen in Figures S1-S5 in Supporting information File”? The supporting Information file is missing. Please introduce in the manuscript the mentioned SEM images and the corresponding discussions in the manuscript, not in the Supplementary information file. I consider that the paper can not be published without microscopic images.
Moreover, the Conclusions chapter should be a little changed. In this chapter there are no summary of all significant information obtained by the Authors and written in the Results. Sentences like” Still, there was a slight change in crystallinity that was evidenced quantitatively by the differences between the values of width peak (FWHM) of samples” makes no sense.
The manuscript is full of grammatical and writing (flow) mistakes. So, the language is not suitable for publication in a scientific journal in many parts. If it possible, I recommend that the authors ask for the help of a native speaker scientist/colleague, who can help with English grammar and scientific writing style.
Finally, I consider that the paper is STILL not proper for publication in the present format and requires large, large revision. I hope that in the near future all the issues will be solved. If the manuscript will not be considerable improved, I will not recommend its publication.
Author Response
This manuscript was somewhat improved after the first submission, but critical mistakes remain present.
Results and discussion section must be considerably improved/ more technically presented. All the figures should be arranged in proper sequence and figure caption should be elaborated. All the results obtained should be compared with the reports (preferably recent) in the literature. More care should be taken to present the results and facilitate understanding of the work.
Author’s comments: We appreciate this comment and accept it. Both sections have been thoroughly revised by authors and English native speakers. All figures were rearranged according to reviewer’s comment. Finally, the references have been updated as well.
“Detailed SEM-EDS images of all samples can be seen in Figures S1-S5 in Supporting information File”? The supporting Information file is missing. Please introduce in the manuscript the mentioned SEM images and the corresponding discussions in the manuscript, not in the Supplementary information file. I consider that the paper can not be published without microscopic images.
Authors’ comments: : We appreciate this comment and accept it. The supporting information file has been submitted conjointly with the main paper. In addition, SEM images have been incorporated to the section of discussion (New Figure 4).
Moreover, the Conclusions chapter should be a little changed. In this chapter there are no summary of all significant information obtained by the Authors and written in the Results. Sentences like” Still, there was a slight change in crystallinity that was evidenced quantitatively by the differences between the values of width peak (FWHM) of samples” makes no sense.
Authors’ comments: : We appreciate this comment and accept it. The conclusion was revised according to reviewer’s comments. We consider that the updated conclusion draws main points and implications based on the presented experimental results.
The manuscript is full of grammatical and writing (flow) mistakes. So, the language is not suitable for publication in a scientific journal in many parts. If it possible, I recommend that the authors ask for the help of a native speaker scientist/colleague, who can help with English grammar and scientific writing style.
Authors’ comments: : We appreciate this comment and accept it. As before mentioned, all text has been revised by English native speaker.
Finally, I consider that the paper is STILL not proper for publication in the present format and requires large, large revision. I hope that in the near future all the issues will be solved. If the manuscript will not be considerable improved, I will not recommend its publication.

Reviewer 2 Report
The manuscript has been well revised. It is recommended for publication after a minor revision in language.
Author Response
Authors’ comments: : We appreciate this comment and accept it. The article has been revised by English native speaker.
Reviewer 3 Report
The authors corrected properly all the points requested by the reviewer.
Author Response
Authors' comments: We are thankful for your comment
Round 3
Reviewer 1 Report
Based on this revised version of the manuscript, there are still weaknesses in this paper.
The Authors did not explain the novelty and motivation of the work. The main goal of the study was not clearly stated. Why these experiments were conducted? What did the Authors want to achieve? In my opinion, answering these questions will be helpful for readers to understand the data and its usefulness. This information should be added in the last paragraph of the Introduction section.
Moreover, please reformulate expression like "Cao et al. improved...", "Supamathanon and Khabuanchalad prepared...", Hu et al. used...", "Sawa et al. showed...", "Fernandes et al. prepared...", "Viswanadham et al. studied...", etc. and connect the introduction to the importance of the work in a gradual manner.
Summarizing, the manuscript may be interesting but it still needs some improvements before it will be accepted to print in this journal.
Author Response
(x) I would not like to sign my review report
( ) I would like to sign my review report
English language and style
( ) Extensive editing of English language and style required
( ) Moderate English changes required
(x) English language and style are fine/minor spell check required
( ) I don't feel qualified to judge about the English language and style
Response: The manuscript has been improved by revising the entire text.
|
Yes |
Can be improved |
Must be improved |
Not applicable |
|
|
Does the introduction provide sufficient background and include all relevant references? |
( ) |
( ) |
(x) |
( ) |
|
Are all the cited references relevant to the research? |
(x) |
( ) |
( ) |
( ) |
|
Is the research design appropriate? |
(x) |
( ) |
( ) |
( ) |
|
Are the methods adequately described? |
(x) |
( ) |
( ) |
( ) |
|
Are the results clearly presented? |
(x) |
( ) |
( ) |
( ) |
|
Are the conclusions supported by the results? |
(x) |
( ) |
( ) |
( ) |
Response: We appreciate the reviewer for confirmation of the improvement of the revised manuscript in the previous round.
Comments and Suggestions for Authors
Based on this revised version of the manuscript, there are still weaknesses in this paper.
The Authors did not explain the novelty and motivation of the work. The main goal of the study was not clearly stated. Why these experiments were conducted? What did the Authors want to achieve? In my opinion, answering these questions will be helpful for readers to understand the data and its usefulness. This information should be added in the last paragraph of the Introduction section.
Response: We appreciate the reviewer for this valuable comment. We have updated the last paragraph of the manuscript as suggested by the reviewer.
Moreover, please reformulate expression like "Cao et al. improved...", "Supamathanon and Khabuanchalad prepared...", Hu et al. used...", "Sawa et al. showed...", "Fernandes et al. prepared...", "Viswanadham et al. studied...", etc. and connect the introduction to the importance of the work in a gradual manner.
Response: We appreciate the reviewer for this valuable comment. We have reorganized the introduction section of the manuscript.
Summarizing, the manuscript may be interesting but it still needs some improvements before it will be accepted to print in this journal.
Response: We appreciate the reviewer for the recommendation.
Round 4
Reviewer 1 Report
Based on this revised version of the manuscript, unfortunately there are still unanswered questions and weaknesses in this paper.
Please improve the quality of Figures 1 and 4. Also, the scale bar of Figure 4 is not visible.
Figure 3. Please remove "grado".
Figure 6. Absorbance (a.u.) on y scale. 500 is not visible on x-scale.
Please use only three significant numbers, i.e. 1.11, 11.1 and 111. please also apply for Figure 6.
Author Response
Dear Reviewer
My coleagues and I are very thankful for your comments by which our article is now much better.
Please improve the quality of Figures 1 and 4. Also, the scale bar of Figure 4 is not visible.
We accept this comment, therefore figures 1-4 was changed according to reviewer's comments
Figure 3. Please remove "grado".
We accept this comment, therefore the figure 3 was changed according to reviewer's comment
Figure 6. Absorbance (a.u.) on y scale. 500 is not visible on x-scale.
We accept this comment, therefore the figure 6 was changed according to reviewer's comment
Please use only three significant numbers, i.e. 1.11, 11.1 and 111. please also apply for Figure 6.
We accept this comment, therefore the figure 6 was changed according to reviewer's comment